# SARS-CoV-2 in Pregnancy: A Comprehensive Summary of Current Guidelines

**DOI:** 10.3390/jcm9051521

**Published:** 2020-05-18

**Authors:** Kavita Narang, Eniola R. Ibirogba, Amro Elrefaei, Ayssa Teles Abrao Trad, Regan Theiler, Roseli Nomura, Olivier Picone, Mark Kilby, Ramón Escuriet, Anna Suy, Elena Carreras, Gabriele Tonni, Rodrigo Ruano

**Affiliations:** 1Maternal-Fetal Medicine Division, Department of Obstetrics and Gynecology, Mayo Clinic College of Medicine, 200 First Street SW, Rochester, MN 55905, USA; Narang.Kavita@mayo.edu (K.N.); Ibirogba.Eniola@mayo.edu (E.R.I.); Elrefaei.Amro@mayo.edu (A.E.); TelesAbraoTrad.Ayssa@mayo.edu (A.T.A.T.); 2Obstetrics Division, Department of Obstetrics and Gynecology, Mayo Clinic College of Medicine, 200 First Street SW, Rochester, MN 55905, USA; Theiler.Regan@mayo.edu; 3Department of Obstetrics and Gynecology, Escola Paulista de Medicina—Universidade Federal de São Paulo, São Paulo 04021, Brazil; roseli.nomura@hotmail.com; 4Groupe de Recherche sur les Infections pendant la Grossesse (GRIG), CNGOF, Service de Gynécologie-Obstétrique Colombes, Publique-Hôpitaux de Paris, Hôpital Louis Mourier, Université de Paris, Inserm IAME-U1137, 75000 Paris, France; olivier.picone@aphp.fr; 5Fetal Medicine Centre, Birmingham Women’s and Children’s Foundation Trust, College of Medical & Dental Sciences, University of Birmingham, Edgbaston, Birmingham B15 2TT, UK; M.D.KILBY@bham.ac.uk; 6Catalan Health Service, Government of Catalonia, 080028 Barcelona, Spain; rescuriet@gencat.cat; 7Department of Obstetrics and Gynecology, Hospital Universitari Vall d’Hebron, 080028 Barcelona, Spain; asuy@vhebron.net (A.S.); ecarreras@vhebron.net (E.C.); 8Prenatal Diagnostic Division, Department of Obstetrics and Gynecology, AUSL di Reggio Emilia Istituto di Ricerca a Carattere Clinico Scientifico, 42100 Reggio Emilia, Italy; Gabriele.Tonni@ausl.re.it

**Keywords:** coronavirus, COVID-19, pandemic, SARS-CoV-2, pregnancy, guidelines, prevention, screening, perinatal

## Abstract

Since the declaration of the global pandemic of COVID-19 by the World Health Organization on 11 March 2020, we have continued to see a steady rise in the number of patients infected by SARS-CoV-2. However, there is still very limited data on the course and outcomes of this serious infection in a vulnerable population of pregnant patients and their fetuses. International perinatal societies and institutions including SMFM, ACOG, RCOG, ISUOG, CDC, CNGOF, ISS/SIEOG, and CatSalut have released guidelines for the care of these patients. We aim to summarize these current guidelines in a comprehensive review for patients, healthcare workers, and healthcare institutions. We included 15 papers from 10 societies through a literature search of direct review of society’s websites and their journal publications up till 20 April 2020. Recommendations specific to antepartum, intrapartum, and postpartum were abstracted from the publications and summarized into Tables. The summary of guidelines for the management of COVID-19 in pregnancy across different perinatal societies is fairly consistent, with some variation in the strength of recommendations. It is important to recognize that these guidelines are frequently updated, as we continue to learn more about the course and impact of COVID-19 in pregnancy.

## 1. Introduction

The World Health Organization (WHO) declared a global pandemic of COVID-19, caused by SARS-CoV-2 on 11 March 2020 [1]. The rapidly escalating numbers of individuals infected globally remain on the rise and little is still known about the course and outcomes of this serious infection in a vulnerable population of pregnant patients and their fetuses.

A variety of professional societies and institutions involved in the care of pregnant patients including Society for Maternal and Fetal Medicine (SMFM) [2,3] from United States, American College of Obstetrics and Gynecology (ACOG) [4,5] from United States, Royal College of Obstetrics and Gynecology (RCOG) [6] from United Kingdom, International Society for Ultrasound in Obstetrics and Gynecology (ISUOG) [7], United States Centers of disease control (CDC) [8,9], the World Health Organization (WHO) [10], College National de Gynecologie et Obstetrique Francais (CNGOF) [11] from France, Istituto Superiore di Sanità/Società Italiana di Ecografia Ostetrico Ginecologica (ISS/SIEOG) [12,13] from Italy, and the Catalan Health Service (CatSalut) [14] from Spain have released independent guidelines for the assessment and care of pregnant patients from prenatal course to intrapartum to postpartum.

A paper published by Boelig et al. in March 2020 [3] to guide Maternal Fetal Medicine specialists on the care of SARS-Cov-2 pregnant patients urged healthcare providers and their institution to develop internal guidelines to have their unit ready to care for these patients.

In order to help institutions keep up with this rapidly evolving landscape, the authors of this paper aim to summarize and discuss all the current guidelines put forth by the aforementioned professional societies and institutions into one document. The goal is to allow institutions access to a comprehensive summary of guidelines related to the SARS-Cov-2 pandemic in pregnancies, which they can adapt to their practice environment and capabilities. The primary focus of all published guidelines is to design a model where patients and their families, as well as healthcare workers (HCW) in the frontline of the pandemic are protected and prepared.

## 2. Experimental Section

Perinatal guidelines that are frequently cited in the United States include publications by SMFM, ACOG, and ISUOG. However, to encompass a global picture and include guidelines that can be generalized to a larger patient population, we included some international guidelines from five countries (US, UK, Italy, Spain, and France). These were selected based on our collaboration with the co-authors from the respective countries to help with translation of documents or clinical application relevance; these societies include RCOG from United Kingdom, CNGOF from France, CatSalut from Spain, and ISS/SIEOG from Italy. Publications from outside of United States were selected by authors affiliated with that country, respectively. We also included WHO and CDC for their expertise in global health care and infectious diseases, respectively. 

A literature search was performed through direct review of all the aforementioned society’s website and journal publications and PubMed. Guidelines published between December 2019 and 20 April 2020, with selection of the most updated versions, were included. The search plan for SMFM, ACOG, and RCOG were arranged and done by two authors K.N. and E.I. with input from the study’s principal investigator—R.R. RCOG publication was reviewed by author M.K. from the UK, CNFOG was reviewed, selected and translated by author O.P., CatSalut publication was reviewed, selected, and translated by authors R.E., A.S., E.C. from Spain, and ISS/SEIOG was reviewed, selected and translated by author G.T. from Italy—all applying the Preferred Reporting Items for Systematic Reviews and Meta Analyses (PRISMA) guidelines for the data extraction and quality assessment. Keywords used to search include COVID-19 and/or SARS-CoV-2 infection in pregnancy and the name of the society. Publications were included if they are an original document from the society, if they outline details on management of patients either during antepartum, intrapartum or postpartum, and if they were expert opinions or expert guidance. Exclusion criteria include case series, case reports, retrospective cohort studies, systematic reviews, or metanalyses. ACOG, SMFM, ISS, and CDC had two publications relevant to guidance for perinatal care and were all included. All other societies had one publication each. A total of 15 papers were identified from 10 societies and reviewed by two authors (K.N. and E.I.) who were in agreement. The list of publications included are summarized in (Table 1), arranged in ascending order of publication date.

All the publications were thoroughly reviewed and important points of discussions were abstracted and summarized into antepartum, intrapartum, and postpartum management, as discussed in the next section. Not all publications addressed every aspect of care, but we summarized the most salient points in each publication in order to highlight the similarities and differences amongst these international guidelines.

## 3. Results

After reviewing all the publications, information was classified into antepartum, intrapartum, or postpartum management and was summarized in a systematic fashion into Table 2, Table 3 and Table 4, respectively.
Prenatal and antepartum care (See Table 2): Reviewed guidelines support some form of screening of pregnant patients depending on symptoms and exposure, use of telehealth is encouraged for prenatal visits, while limiting face to face visits and ultrasounds only to those that are medically necessary. Prenatal appointments, lab work, and ultrasounds should be scheduled on the same day if possible. All ultrasound equipment and patient rooms should be appropriately cleaned after each use. The use of antenatal corticosteroids for fetal lung maturation can be continued till 34 weeks gestation, but the use of steroids in the late preterm period, >34 0/7 weeks gestation remains controversial.Intrapartum care (See Table 3): Reviewed guidelines recommend a designated area within the unit to care for SARS-CoV-2 positive pregnant patients or Person under investigation (PUI). Timing and mode of delivery should follow routine obstetric indications. Cesarean section (CS) should be reserved for obstetric indications only; infection with SARS-CoV-2 is not an indication for cesarean delivery unless there is acute decompensation of mother or fetus. Only one consistent asymptomatic support person is allowed to be present at time of delivery. Patients and healthcare workers should be appropriately gowned, gloved, and have protective face masks; specifically, N95 should be used for aerosol generating procedures such as forceful expiration during pushing, use of oxygen for intrauterine resuscitation, or intubation. Use of operative delivery to shorten the second stage of labor can be considered for routine obstetric indications. There is no contraindication to regional or general anesthesia if indicated, but appropriate personal protective equipment (PPE) use is encouraged.Postpartum care (See Table 4): Reviewed guidelines encourage early discharge from the hospital, one day for vaginal delivery and two days for cesarean delivery. This limits face to face exposure and increases bed availability. Separation of mother and baby or discouraging breastfeeding are not advised, unless the mother is acutely ill. However, mothers are encouraged to (1) practice respiratory hygiene during feeding, (2) wear a mask, (3) wash hands before and after touching the baby, and (4) routinely clean and disinfect surfaces they have touched. If breastpumping is used, all equipment should be cleaned thoroughly before and after each use. Postpartum visits should be performed over telehealth, unless face to face visit is essential to management.

## 4. Discussion

The summary of the reviewed guidelines for the management of COVID-19 in pregnancy across different professional societies and institutions is consistent, with some variation in the strength of recommendations. Global societies such as WHO and CDC have a similar approach to their guideline publication, keeping their recommendations broad so it can be utilized across all shapes and sizes of healthcare institutions. Many of their recommendations overlap with those for the general population and they provide great resources to guide readers to perinatal societies for more specific questions.

International perinatal societies, including ACOG, RCOG, SMFM, ISUOG, CNGOF, ISS/SIEOG, and public institution CatSalut, all share similar recommendations answering questions that are very specific to the care of pregnant patients—from prenatal screening, antepartum care, details of intrapartum care during different stages of labor in emergency and non-emergency settings to postpartum care and follow up. The guidelines put forth by SMFM (United States) are most specific to the care of high risk pregnancies, given their expertise in this field. ACOG (United States) and RCOG (United Kingdom) summarize recommendations that are suitable for lower risk pregnant patients. CNGOF (France) and ISS/SIEOG (Italy) and CatSalut (Barcelona) give some practical recommendations for the management of infected pregnant women. ISUOG (International) provides more information specific to managing and cleaning ultrasound equipment—an essential tool in the care of pregnant patients, which could be a vector for disease transmission if sanitization is not a priority.

The consensus amongst all perinatal societies encourages all institutions to transition to telehealth when appropriate and limit the number of face to face visits. Ultrasounds and antenatal surveillance should be performed only if medically indicated. The use of antenatal steroids for fetal lung maturation for patients at high risk of preterm birth within seven days should still be performed if pregnancy is between 24 0/7 to 33 6/7 weeks gestation, but use during late preterm of 34 0/7 to 36 6/7 weeks gestation is still controversial. All institutions should set up a designated screening area, labor and delivery rooms, and operating rooms for SARS-CoV-2 infected patients. All patients should be screened for symptoms, travel history, contact history, and follow the appropriate algorithm provided to guide need for performing real time PCR tests. If elective procedures or induction of labor is scheduled, patients should first be screened and triaged over the phone, followed by a nasopharyngeal swab for SARS-CoV-2 infection. This screening should be done within a time frame to allow the test results to return before the scheduled procedure date. For urgent or emergent obstetric conditions, screening for SARS-CoV-2 should be performed right away, but procedures should not be delayed for results to return; patients should be treated as a PUI and managed as presumptive positive.

As the numbers of testing sites and resources have increased over the past few weeks, there should be consideration for screening every pregnant patient being admitted, regardless of exposure, history or symptoms. Societies recommend only one consistent support person to be present during delivery. Mode and timing of delivery should still be performed on the basis of routine obstetric indications, and delivery should be expedited with cesarean delivery in the event of maternal deterioration due to severe COVID-19 disease or fetal distress. Aerosol generating procedures such as the use of supplemental oxygen, intubation, and forceful pushing should be avoided to protect everyone in the delivery room. Appropriate PPE should be donned by patients and healthcare workers during all interactions. N95 should be worn during aerosol generating procedures.

Currently, there is no definitive evidence to suggest vertical transmission of SARS-CoV-2. As a result, mother and baby separation and discouraging breastfeeding are not advised unless the mother is acutely ill. Mothers who are acutely ill with SARS-CoV-2 infection are advised the option for breast pumping, and to wash hands before handling baby or touching pumps or bottle, avoid coughing while baby is feeding, and consider wearing a face mask while feeding or handling baby. If a breast pump is used, clean properly after each use and routinely clean all surfaces that are touched. The length of hospital stay should be decreased to one day for vaginal delivery and two days for cesarean delivery to limit time of exposure for patients and healthcare workers in the hospital while also increasing bed capacity. Once discharged, patients are advised to continue social distancing, and routine postpartum visits can be conducted using telehealth. The method of telehealth should be individualized based on institution resources and availability.

## 5. Conclusions

The present manuscript summarizes the guidelines for Obstetrical and perinatal management of pregnant women during the SARS-CoV-2 pandemic, which can be an overall reference for Obstetricians all over the world. Many similarities are identified amongst these guidelines. All of the international professional societies and institutions discussed in this paper, including ACOG, RCOG, SMFM, ISUOG, WHO, CNGOF, ISS/SIEOG, CatSalut and CDC, continue to work tirelessly to put forth updated information for the care of pregnant patients and beyond. This manuscript also provides the summary of the source for continuous updates. It is imperative for readers to continue to use the most updated guidelines available as we continue to learn more about the impacts of the SARS-Cov-2 pandemic in pregnancy.

## Figures and Tables

**Table 1 jcm-09-01521-t001:** Sources of evidence.

Journal/Website	Professional Society/Institution	Publication Title	Publication Date
ISS website https://www.iss.it/coronavirus [12]	Superior Institute of Health	Educational course on Health Emergency on novel Coronavirus	28 February 2020
UOG Journal https://obgyn.onlinelibrary.wiley.com/doi/10.1002/uog.22013 [7]	ISUOG (International Society)	ISUOG Interim Guidance on 2019 novel coronavirus infection during pregnancy and puerperium: information for healthcare professionals	12 March 2020
ACOG website https://www.acog.org/clinical/clinical-guidance/practice-advisory/articles/2020/03/novel-coronavirus-2019 [4]	ACOG (United States)	Novel Coronavirus 2019- Practice advisory	13 March 2020
CDC website https://www.cdc.gov/coronavirus/2019-ncov/hcp/index.html [9]	CDC (United States)	Information for Healthcare Providers: COVID-19 and Pregnant women	16 March 2020
WHO website https://www.who.int/reproductivehealth/publications/maternal_perinatal_health/anc-positive-pregnancy-experience/en/ [10]	WHO (International Society)	Q&A on COVID-19, pregnancy, childbirth, and breastfeeding	18 March 2020
AJOG-MFM Journal https://www.sciencedirect.com/science/article/pii/S2589933320300367 [3]	SMFM (United States)	MFM Guidance for COVID-19	19 March 2020
CNGOF https://pubmed.ncbi.nlm.nih.gov/32199996/ [11]	CNGOF (France)	SARS-CoV-2 infection during pregnancy. Information and proposal of management care. CNGOF	19 March 2020
CatSalut Website https://canalsalut.gencat.cat/ca/salut-a-z/c/coronavirus-2019-ncov/professionals/consulta/?cat=8460bdf4-691a-11ea-88fa-005056924a59&submit=true [14]	CatSalut (Barcelona)	SARS-CoV-2 coronavirus infection Information for pregnant women and their families.	20 March 2020
ACOG website https://www.acog.org/clinical-information/physician-faqs/covid-19-faqs-for-ob-gyns-obstetrics [5]	ACOG (United States)	COVID-19 FAQs for Obstetrician-Gynecologists, Obstetrics	23 March 2020
AJOG-MFM Journal https://www.sciencedirect.com/science/article/pii/S2589933320300409 [2]	SMFM (United States)	Labor and Delivery Guidance for COVID-19	25 March 2020
ISS website https://www.iss.it/coronavirus [12]	Superior Institute of Health (Italy)	Rational use of individual protection devices in the assistance of Covid-19 patients	28 March 2020
SIEOG website https://www.sieog.it/events/emergenza-covid19/ [13]	Italian Society for Ultrasound in Obstetrics and Gynecology (Italy)	SARS-Cov-2 Pandemic: Information and Recommendation	29 March 2020
CDC website https://www.cdc.gov/coronavirus/2019-ncov/need-extra-precautions/pregnancy-breastfeeding.html [8]	CDC (United States)	COVID-19: Pregnancy and breastfeeding	3 April 2020
CatSalut Website https://canalsalut.gencat.cat/ca/salut-a-z/c/coronavirus-2019-ncov/professionals/consulta/?cat=8460bdf4-691a-11ea-88fa-005056924a59&submit=true [14]	CatSalut (Barcelona)	Clinical guideline for new cases of SARS-CoV-2 coronavirus infection in pregnant women and infants	6 April 2020
RCOG Journal https://www.rcog.org.uk/globalassets/documents/guidelines/2020-04-17-coronavirus-covid-19-infection-in-pregnancy.pdf [6]	RCOG (United Kingdom)	COVID-19 infection in pregnancy	17 April 2020

**Table 2 jcm-09-01521-t002:** Summary of guidelines for antepartum care of pregnant patients during the COVID-19 pandemic.

TITLE	Professional Society	ISUOG	CNGOF	ACOG	SMFM	RCOG	WHO	CDC	CatSalut	ISS/SIEOG
PRENATAL CARE AND ANTEPARTUM	Infection Screening	Set up triage screening area. All outpatients should be assessed and screened for TOCC and symptoms. All HCW should wear appropriate PPE. All suspected cases should be screened with qRT-PCR. Repeat testing in 24 hr if negative, but still high suspicion. Chest CT should be considered, if high suspicion	Set up triage screening area Patients should be provided with a surgical mask at the entrance (and is not to be removed until the patient is isolated in a suitable room). All outpatients should be assessed and screened for TOCC and symptoms. All HCW should wear appropriate PPE. All suspected cases should be screened with qRT-PCR. Symptomatic patients should be treated as positive till results are back Patients with suspected COVID-19 who present with obstetric emergency should be transferred immediately to an isolation room by HCW using appropriate PPE. Obstetric management should not be delayed for COVID-19 testing	Routine screening before appointment, if suspicious. If symptomatic, initiate testing and notify health department, mark patient as PUI. Screening algorithm https://www.acog.org/-/media/project/acog/acogorg/files/pdfs/clinical-guidance/practice-advisory/covid-19-algorithm.pdf. All HCW should wear PPE (Face mask, Eye protection, gloves, and gown)	Triage symptomatic patients via telehealth. Test anyone with new flu-like symptoms, especially older, immune-compromised, advanced HIV, homeless, hemodialysis. Utilize drive through or standalone testing area. Symptomatic patients should be treated as positive till results are back	Patients should be provided with a surgical mask at the entrance (and is not to be removed until the patient is isolated in a suitable room). Screen all patients presenting to maternity unit. Patients who meet criteria (see guideline for details) for potential COVID-19should have a full blood count evaluation; if lymphopenia is identified, COVID-19 testing should be arranged. Patients with suspected COVID-19 who present with obstetric emergency should be transferred immediately to an isolation room by HCW using appropriate PPE. Obstetric management should not be delayed for COVID-19 testing Symptomatic patients should be treated as positive till results are back	-Testing protocols and eligibility vary depending on where you live. Symptomatic and high risk patients should get screening priority. HCW should maintain hand hygiene, and appropriate use of protective clothing like gloves, gown, and medical mask.	Same as general population	Women are asked to phone prior to antenatal visit. All women should take preventive measures when attending health care settings. Screen women with symptoms presenting to antenatal clinic. Symptomatic patients should be treated as positive till negative results are back	Telephone triage. Screen with symptoms Asymptomatic mothers: respect hygiene measures, social distancing. PPE not required. Symptomatic mothers: individual PPE required for mothers and HCW. Symptomatic mothers are tested for SARS-CoV-2 using nasopharyngeal swabs and isolation in a dedicated room. Public Hygiene Service should be informed.
	Place of care	Negative pressure or single isolation rooms in tertiary care center. Reserve ICU for critical patients	Isolation room for patients with suspected/confirmed COVID-19 for whom care cannot be safely delayed for self-isolation	N/A	Designated COVID-19 area within the facility	Isolation room for patients with suspected/confirmed COVID-19 for whom care cannot be safely delayed for self-isolation	−N/A	Same as general population	Same as general population. If hospital admission is needed, women are referred to one reference hospital in the Region	Asymptomatic: delivery at General Hospitals. Stable symptomatic mothers delivery at General hospital with designated area within the facility. Delivery in isolated room. Mothers, medical staff and a single accompanying person must wear PPE. Room ventilation at least with 60 L/s. Mothers & babies are kept in isolated room. Unstable symptomatic mothers delivery at tertiary care center with ICU facilities
	Prenatal appointment	Postpone by 14 days if positive or until 2 negative results	Elective and non-urgent appointments should be postponed or completed by telehealth. Encourage use of telehealth for all visits–HCW meetings should all be virtual/audio. Keep some providers at home. No support persona at outpatient visit	If, after screening, the patient reports symptoms of or exposure to a person with COVID-19, that patient should be instructed not to come to the health care facility for their appointment and health care clinicians should contact the local or state health department to report the patient as a possible person under investigation (PUI)	Elective and non-urgent appointments should be postponed or completed by telehealth. Encourage use of telehealth for all visits. HCW meetings should all be virtual/audio. Keep some providers at home. No support persona at outpatient visit. Labs and US at the same appointment. Provide patient with ambulatory BP cuff/machine. F2F visits at 11-13,20,28,36 weeks and weekly after 37 weeks	-Routine appointments for women with suspected/confirmed COVID 19 should be delayed until after the recommender period of self-isolation. For symptomatic patients, defer appointments until 7 days after symptom onset; defer appointments for 14 days for patients with symptomatic household contacts. Encourage the use of telephone for non-urgent consultation/enquiries	N/A	N/A	Encourage use of telehealth for all visits	Routine antenatal care appointment monthly for asymptomatic mothers. Planned visit @ at maternity unit at 37-38 weeks and then at 40 weeks, if physiologic pregnancy. Symptomatic mothers: delay appointments for several days according to symptoms, recommend GP consultation and keep telephone contacts
	**Ultrasound frequency**	Suspected, asymptomatic confirmed and recovering patients: US q 2–4 weeks for Fetal growth and AFI, UA dopplers if indicated	Continue US as medically indicated when possible.	Continue US as medically indicated when possible. Elective US should not be performed. Postpone or cancel testing or examinations if the risk of exposure and infection within the community outweighs the benefit of testing.	Combine dating and NT in 1st trimester. Anatomy scan at 20–22 weeks. Consider stopping serial CL after anatomy US if TVUS CL ≥35 mm, prior preterm birth at >34 Weeks. BMI >40: schedule at 22 weeks to reduce risk of suboptimal views/need for follow up. Single growth F/U at 32 weeks. Low lying placenta F/U 34–36 wks. Refer to primary publication for disease specific US frequency	In addition to routine ultrasound surveillance, fetal growth restriction surveillance is recommended 14 days after resolution of acute illness due to theoretical risk of growth restriction.	N/A	N/A	Continue US as medically indicated when possible	Asymptomatic mothers: continue US assessment as routine. Symptomatic mothers: following 14 days of isolations and resolution of symptoms, general clinical examination and ultrasound assessment for fetal growth every 3–4 weeks
	**Ultrasound Equipment/patient rooms**	Must be cleaned with disinfectant per manufacturer guidelines after EVERY use Deep clean of all instruments and room in case of positive patient	Must be cleaned with disinfectant per manufacturer guidelines after EVERY use	N/A	Wipe down patient rooms after every visit of suspected SARS-COV-2 patients	-Decontaminate after use on suspected or confirmed SARS-CoV-2 patients	N/A	N/A	Deep clean of all instruments and room in case of positive patient	Deep clean of all instruments and room ventilation every 10 min
	**Antenatal corticosteroids** **(BMZ)**	Avoid in critically ill patient; risk of worsening disease	Should continue if <34 weeks, even if tested positive for SARS-CoV-2	Should continue if <34 weeks, even if tested positive for COVID-19 Controversial for 34 0/7–36 6/7 Weeks Other modifications should be individualized	Judicious use <34 weeks Avoid >34 weeks	Administer for routine indications No evidence to suggest harm in the context of SARS-CoV-2 infection	N/A	N/A	Administer for routine indications	N/A
	**GBS screening**	Delay by 14days in patients with TOCC risk factors	As indicated between, 36 0/7–37 6/7 weeks gestation.	As indicated between, 36 0/7–37 6/7 weeks gestation. Patients can self-collect with proper instructions if the resources and infrastructure allow	Routine screening at 36wks	N/A	−N/A	N/A	Routine screening	Routine screening
	**Antenatal surveillance (BPP, NST)**	N/A	Daily NST if patient hospitalized	Reserve for medically indicated screening During acute illness, fetal management should be similar to that provided to any ill pregnant person.	Limit NSTs if <32 wks Twice weekly NST only for FGR with abnormal UA Doppler studies If patient needs US, perform BPP instead of NST Kick counts instead of NST for low risk patients	N/A	N/A	N/A	N/A	N/A

Travel history, occupation, significant contact and cluster (TOCC), Healthcare worker (HCW), Personal protective equipment (PPE), Face to face (F2F), General practitioner (GP), Nuchal Translucency (NT), Follow up (F/U), Ultrasound (UA), fetal growth restriction (FGR), Non-stress test (NST), Biophysical profile (BPP).

**Table 3 jcm-09-01521-t003:** Summary of guidelines for intrapartum care of pregnant patients during the COVID-19 pandemic.

TITLE	Professional Society	ISUOG	CNGOF	ACOG	SMFM	RCOG	WHO	CDC	CatSalut	ISS/SIEOG
INTRAPARTUM CARE	**Pre-Delivery preparation**	Social distancing	Social distancing	Obstetric, pediatric or family medicine, and anesthesia teams should be notified in order to facilitate care.	Social distancing and off work for 2 weeks prior to anticipated delivery (start at ~37wks). Screen patient and partner on phone day before admission. Institution should run simulations	Minimum staffing and social distancing. Screen patient and partner at maternity unit. Partners with symptoms less than 7 days prior should be instructed to self-isolate and not be allowed into the maternity unit. Women with suspected or confirmed COVID19 should be encouraged to remain at home during early labor; women in active labor should be admitted to an isolation room. Dry run simulations for elective/emergency procedures	If COVID19 is suspected or confirmed, health workers should take all appropriate precautions to reduce risks of infection to themselves and others, including hand hygiene, and appropriate use of protective clothing like gloves, gown and medical mask.	N/A	Screen women with symptoms at maternity unit Respectful care must be always considered minimum number of professionals attending women. Only one partner allowed for companionship during labor and delivery.	Screen women with symptoms at maternity unit. Asymptomatic: as per routine care. Stable Symptomatic patients admit to hospital. Unstable symptomatic patients: refer to hospitals with ICU facility
	**Delivery Time**	Based on routine obstetric indications Early delivery should be considered for critically ill patients	If infection in early pregnancy with recovery, No alterations in delivery time. Based on routine obstetric indications Early delivery should be considered for critically ill patients	If infection in early pregnancy with recovery, No alterations in delivery time. If infection in late pregnancy and recovery, postpone delivery (if no other medical indications arise) until a negative testing result is obtained or quarantine status is lifted in an attempt to avoid transmission to the neonate. COVID-19 is not an indication of delivery.	Based on routine indications. No contraindication to induction of labor unless beds are limited. For term COVID-19 patients, consider delivery because symptoms peak in 1–2 weeks after onset	Based on routine indications	N/A	N/A	PPE in all cases. Continuous fetal electronic monitoring.	Per routine Obstetric indications
	**Delivery location**	Designated negative pressure isolation room	Designated isolation room, for suspected or confirmed cases of COVID-19	N/A	Designated delivery and operating rooms	Designated isolation room, for suspected or confirmed cases	N/A	N/A	Designed isolation room for suspected or confirmed cases of COVID-19. Designed negative pressure isolation room for CS	Designated isolated room for suspected or confirmed cases
	**Mode of Delivery**	Based on routine obstetric indications Infection is NOT an indication for CS. Expedite delivery by CS in setting of fetal distress or maternal deterioration. Water birth should be avoided	Based on routine obstetric indications Infection is NOT an indication for CS. Expedite delivery by CS in setting of fetal distress or maternal deterioration	Per routine obstetric indications. No specific recommendations for CS. Operative vaginal delivery is not indicated for suspected or confirmed cases alone, but can be used as routinely indicated	Based on routine obstetric indications. Infection is NOT an indication for CS	Based on routine obstetric indications unless maternal respiratory condition demands early delivery. Water birth should be avoided.	As clinically indicated	N/A	Based on routine obstetric indications. Infection is not an indication for CS	Routine obstetric indications. Infection is not an indication for CS
	**Support person**	Limit visitors, no clear number specified	No visitor	Allowed one consistent asymptomatic support person	Allowed one consistent support person. No children <16–18 y/o	Allowed one consistent asymptomatic support person who should be restricted to the patient’s bedside.	N/A	N/A	Companionship by one person relative to the women is encouraged during all the labor and delivery	Single accompanying asymptomatic person
	**Obstetric Analgesia and Anesthesia**	Regional anesthesia and GA can be considered	Regional anesthesia and GA can be considered	N/A	Avoid use of nitrous oxide	NO evidence against regional or GA. Epidural analgesia is recommended in suspected or confirmed cases, to minimize the need for GA if urgent delivery is needed.	N/A	N/A	Epidural analgesia is recommended to women with suspected or confirmed COVID-19 to minimize the need for GA if urgent delivery is needed.	Regional and GA can be considered
	**Second Stage of Labor**	Consider shortening with operative delivery to minimize aerosolization and maternal respiratory effort	N/A	N/A	Do not delay pushing. Considered aerosolizing, N95 should be worn by HCW and patients	Consider shortening with operative vaginal delivery in symptomatic women who become exhausted or hypoxic	N/A	N/A	N/A	N/A
	**Third stage of Labor**		N/A	N/A	Active management to reduce blood loss (national blood shortage)	N/A	N/A	N/A	Active management in all cases	Per routine
	**Oxygen supplementation**	N/A	N/A	N/A	Considered aerosolizing, HCW must wear appropriate PPE. Do not use O_2_ for intrauterine resuscitation	Hourly O_2_ sat measurements (in addition to routine maternal-fetal observations) for women with suspected/confirmed COVID-19. Aim to keep o_2_ sat >94%, titrating O_2_ therapy accordingly.	N/A	N/A	N/A	N/A
	**Umbilical cord clamping**	Avoid delayed cord clamping in confirmed and suspected cases	N/A	No recommendations against delayed clamping of Umbilical cord.	Avoid delayed cord clamping	Delayed cord clamping is still recommended in the absence of contraindications	N/A	N/A	Delayed cord clamping is still recommended in the absence of contraindications	General rule
	**PPE use**		N/A	N/A	Asymptomatic or negative patients Patient and provider wear surgical mask. Aerosolizing procedures-N95 for patient and N95, gown, gloves, face shield for provider	Level of PPE should be based on the risk of requiring GA. Aerosolizing procedures-use FFP3 mask	N/A	Same as general population	N/A	Symptomatic with stable or unstable condition: Mothers, medical staff and accompanying person must wear all protection devices. Masks should be FFP2/FFP3 type.
	**Elective Cesarean delivery/induction of labor (IOL)**		N/A	N/A	No contraindication to IOL unless there is limited beds	For suspected/confirmed cases, consider delay of elective CD or IOL if safely feasible to	N/A	N/A	N/A	N/A

General anesthesia (GA), Quantitative real time polymerase chain reaction (qRT-PCR), Ultrasound (US), Umbilical artery (UA), Biophysical profile (BPP), Non stress test (NST), Cesarean Section (CS).

**Table 4 jcm-09-01521-t004:** Summary of guidelines for postpartum care of pregnant patients during the COVID-19 pandemic.

TITLE	Professional Society	ISUOG	CNOGF	ACOG	SMFM	RCOG	WHO	CDC	CatSalut	ISS/SIEOG
POSTPARTUM CARE	Placental or fetal tissue	Should be handled as infectious tissue in positive patients Consider qRT-PCR on placenta	N/A	N/A	N/A	N/A	N/A	N/A	N/A	N/A
	Length of stay	N/A	N/A	Expedited discharge should be considered. VD—1 day CS—2 days	Expedited discharge should be considered. VD—1 day CS—2 days	N/A	N/A	Same as general population	Expedited discharge should be considered	Asymptomatic→2 days Symptomatic→3 days
	Breastfeeding	Insufficient evidence Okay for asymptomatic patients, mothers should use masks and wash hands Separation and breast pumping suggested in critically ill patients	Limited evidence to advise against breastfeeding. Advise patients to: wash hands before handling baby, touching pumps or bottle; avoid coughing while baby is feeding; consider wearing face mask while feeding or handling baby;	N/A	Advice patients to: wash hands before handling baby, touching pumps or bottle; avoid coughing while baby is feeding; consider wearing face mask while feeding or handling baby; if breast pump us used, clean properly after each use; consider asking someone who is well to feed baby.	No contradictions Advice patients to: wash hands before handling baby, touching pumps or bottle; avoid coughing while baby is feeding; consider wearing face mask while feeding or handling baby; if breast pump us used, clean properly after each use; consider asking someone who is well to feed baby.	Women with COVID-19 can breastfeed if they wish to do so. They should: (1) Practice respiratory hygiene during feeding (2) wear a mask (3) Wash hands before and after touching the baby (4) Routinely clean and disinfect surfaces they have touched.	During separation encourage dedicated breast pump. Mother should use a facemask and practice hand hygiene after each feeding	Encourage breastfeeding support (1) Practice respiratory hygiene during breastfeeding. (2) wear a mask (3) hands and tissues hygiene before and after breastfeeding	Encourage breastfeeding support. Symptomatic: (1) hands and tissue hygiene (2) wear a surgical mask
	Skin to skin	Can be considered with appropriate PPE use for asymptomatic patients	N/A	N/A	N/A	Routine precautionary separation of a healthy baby and mother is not advised at this point.	Allow with precautions and good hygiene clean.	N/A	Individualize according to the conditions of the mother and the baby	N/A
	Postpartum pain control	N/A	N/A	N/A	No contraindication to NSAID use	N/A	N/A	N/A	N/A	N/A
	Postpartum visit	N/A	Encourage telehealth for postpartum visit	Encourage telehealth for postpartum visit. Delay comprehensive face to face postpartum visit to 12 weeks. Use telehealth before 12 weeks.	Encourage telehealth for postpartum visit	Encourage telehealth for postpartum visit	N/A	N/A	Stay at home policy and encourage of telehealth postpartum visits home visit by health professional (midwife) between 48–72 h after discharge	N/A

Vaginal Delivery (VD), Cesarean Section (CS).

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
