# Peer review of "SARS-CoV-2 in Pregnancy: A Comprehensive Summary of Current Guidelines"

_jcm, 2020, doi:10.3390/jcm9051521_

Round 1

Reviewer 1 Report

General

I had the opportunity to review the manuscript titled “SARS-CoV-2 in Pregnancy: A comprehensive summary of current guidelines“ by Narang et al. The aim of this studie was to review ssveral guidelines concercing SARS-CoV-2 and pregnancy.
As there is poor evidence of risks and complications of COVID-19 in pregnancies and there are several guidelines published by a variety of societies this is an important review for involved persons.

  • In general, please pay attention to correct naming of infection (SARS-CoV-2) and disease (COVID-19) caused by this virus:
    It’s correct to write: Infection with SARS-CoV-2, not „infection with COVID-19“ – please review the whole manuscript on the correct nomenclature!
    A patient is SARS-CoV-2 positive, but not COVID-19 positive!
  • Please review spelling as there are some mistakes in the manuscript.

Title

No specific comments. Well chosen title.

Abstract

The abstract is sound and proof. No specific comments.

Keywords

No specific comments.

Introduction

Introduction leads well to the topic of the manuscript.

Experimental section

I was wondering why German speaking recommendations are missing. I suggest to add german guidelines to your review.

Results

Different guidelines are compared in tables 2,3,4. See comments about table structure below.
Please add german recommendations.

Discussion

Discussion is well written and summarizes overlaps and deviations of several guidelines.

Tables and Figures

  • Please introduce all abbreviations (e.g. Table 2: HCW?, PPE?) and provide them under the tables
  • The structure of tables should be revised to simplify reading as these tables are core components of this manuscript

Conclusion

No specific comments

References

No specific comments

Author Response

Comments from Reviewer 1

General- I had the opportunity to review the manuscript titled “SARS-CoV-2 in Pregnancy: A comprehensive summary of current guidelines by Narang et al. The aim of this study was to review several guidelines concerning SARS-CoV-2 and pregnancy.
As there is poor evidence of risks and complications of COVID-19 in pregnancies and there are several guidelines published by a variety of societies this is an important review for involved persons.

    1. In general, please pay attention to correct naming of infection (SARS-CoV-2) and disease (COVID-19) caused by this virus: It’s correct to write: Infection with SARS-CoV-2, not “infection with COVID-19“– please review the whole manuscript on the correct nomenclature! A patient is SARS-CoV-2 positive, but not COVID-19 positive.
      Response: The article has been reviewed and these changes have been made. See tracked changes.
    2. Please review spelling as there are some mistakes in the manuscript.
      Response: The article has been reviewed and spelling errors were corrected. See tracked changes- Page 2- Lines 51, 77, 89, Page 4- Lines 102, 110, 117, 122, Page 5-Lines 125, 126, Page 19-Line 184. We changed the use of the word Cesarean delivery (CD) to Cesarean Section (CS)- See page 4, Line 112 (and in tables), Page 19- Line 140

    3. Title- No specific comments. Well-chosen title.
    4. Abstract- The abstract is sound and proof. No specific comments.
    5. Keywords- No specific comments.
    6. Introduction- Introduction leads well to the topic of the manuscript.
    7. Experimental section- I was wondering why German speaking recommendations are missing. I suggest adding German guidelines to your review.
      Response: Thank you for this great suggestion and we would love to include it. However, at this time, we do not have an author to help us with the translation of the German guidelines.

    8. Results: Different guidelines are compared in tables 2, 3, 4. See comments about table structure below.
    9. Discussion: Discussion is well written and summarizes overlaps and deviations of several guidelines.
    10. Tables and Figures- Please introduce all abbreviations (e.g. Table 2: HCW?, PPE?) and provide them under the tables
      Response: This has been added to the bottom of each table- See Page 11- Line 132 and bottom of page 15.The manuscript has also been revised in the paragraphs to make sure that abbreviations are added when the words were first used- See Page 2- Line 63, Page 4-Line 119.
      References for SIEOG in Table 1 was updated, Page 4.The structure of tables should be revised to simplify reading as these tables are core components of this manuscript
      Response: We reviewed the table to delete redundant words or phrases to shorten the content and thus, making it more legible. Please kindly see tracked changes within all three tables.
    11. Conclusion- No specific comments
    12. References- No specific comments
      Response: We did review the references again to update the format for references 8 and 9

Reviewer 2 Report

Authors report a good review of available guidelines to manage COVID19 In pregnant women.

All issues are well summared in text and tables.

in order to improve Manuscript i’ll Add a specific paragraph regarding the different triage selection for elective or urgent access tempo hospital for pregnant women. This is a fundamental point of daily management for patients and also their first degree relative.

Author Response

Comments from Reviewer 2

Authors report a good review of available guidelines to manage COVID19 In pregnant women. All issues are well summarized in text and tables. In order to improve Manuscript, I’d add a specific paragraph regarding the different triage selection for elective or urgent access tempo hospital for pregnant women. This is a fundamental point of daily management for patients and also their first degree relative.

Response: These changes have been made and a paragraph has been added on Page 19- Line 165 to 170- “ If elective procedures or induction of labor is scheduled, patients should first be screened and triaged over the phone, followed by a nasopharyngeal swab for SARS-CoV-2 infection. This screening should be done within a time frame to allow the test results to return before the scheduled procedure date. For urgent or emergent obstetric conditions, screening for SARS-CoV-2 should be performed right away, but procedures should not be delayed for results to return; patients should be treated as a PUI and managed as presumptive positive.”